# Understanding Teacher Digital Competence in the Framework of Social Sustainability: A Systematic Review

Ana María De la Calle [1,*], Alejandra Pacheco-Costa [2], Miguel Ángel Gómez-Ruiz [3] and Fernando Guzmán-Simón [1]

1. Department of Didactics of Language and Literature and Integrated Philologies, Faculty of Education, Universidad de Sevilla, Pirotecnia s/n., 41013 Sevilla, Spain; fernandoguzman@us.es
2. Department of Arts Education, Faculty of Education, Universidad de Sevilla, Pirotecnia s/n., 41013 Sevilla, Spain; apacheco@us.es
3. Didactics Department, Faculty of Education, Universidad de Cádiz, Avenida República Árabe Saharaui s/n., 11519 Puerto Real, Spain; miguel.gomez@uca.es
* Correspondence: anamariadelacalle@us.es

**Abstract:** Due to the global COVID-19 pandemic, ICT has been urgently introduced in education systems in a generalised manner. In this context, it is essential for teachers to master a spectrum of basic digital competencies and manifest digital leadership in the classroom. In addition, it is necessary to consider the relationship between digital competence development and social sustainability, that is, social and cultural heritage, and to what extent they contribute to improving social cohesion and living conditions in a community. This study presents a systematic review of research on teacher digital competence and social sustainability based on the PRISMA model and a review of 22 studies indexed in SCOPUS. The review reveals that most are intended to measure the digital competence level of teachers, usually in compulsory stages of the educational system and through quantitative studies based on virtual questionnaires comprised of closed-ended questions. However, the studies tend to ignore questions related to social sustainability (access to resources, heritage culture, intergenerational transmission, employability, or gender equality). It is therefore urgent to develop research committed to a sustainable society that is oriented towards social justice.

**Keywords:** digital competence of teachers; social sustainability; systematic review





## 1. Introduction

### 1.1. The Role of Information and Communication Technologies Applied to Education in Today's Society

We are living in times of change where many of the traditional foundations of formal education are being reconceptualized. These modifications to educational practice and new approaches have not been directly promoted by scientific progress in the pedagogical field or by the reflection or innovation of teachers in their professional practice, but by the sudden appearance of a global pandemic caused by a virus, SARS-CoV-2. On the one hand, COVID-19 has generally modified social practices due to the global health emergency it has caused. On the other, this virus has forced the urgent modification of the realities of educational institutions, in many cases with unpredictable consequences [1]. According to data from the United Nations Educational, Scientific, and Cultural Organisation (hereafter, UNESCO), the number of students directly affected at the peak of the pandemic has reached 1,319,558,795 individuals. In fact, it is estimated that one year later, half of the world's students continue to suffer its consequences, as evidenced by the data indicating that more than 100 million children are below the minimum level of reading proficiency as a result of this crisis [2].

Lockdowns and the avoidance of direct contact with other people have led to the search for alternatives that favour personal interactions without the need for face-to-face

contact or manipulation of interchangeable physical formats both at an educational and social level. In this context, Information and Communication Technologies (hereafter, ICT) have definitely come to occupy a central role in our lives. For years, it has been considered that we are immersed in an information and knowledge society, a digital era where, thanks to the use of technologies, distances have been reduced and connectivity to the large internet network is constant [3]. It could be argued that the COVID-19 pandemic has definitely underpinned this omnipresent technological reality that mediates at the personal, social, and organisational levels, and that can even push us to live in dual worlds [4].

In this new panorama, the world's educational systems have found it necessary to introduce ICT, in a generalised manner, in their day-to-day lives as the only means to ensure the continuity of their training activities. The inclusion of ICT in education has caused uncertainty and anxiety not only in families and teachers, but also in the students themselves [5]. Similarly, the extensive and intensive use of ICT has provoked new reflections, if not reluctance, from the teachers who had tried to stay away from the virtual world in their teaching practice up to that point. In some cases, the critical issues go beyond the superficial use of technologies towards the foundations of a traditional system, where the memory-based evaluation of knowledge and the exhaustive control of students' behaviour seemed the only visible and relevant activities of teachers [6].

The changes described above highlight the weaknesses of ICT in education (such as the lack of technical accessibility due to economic reasons or connection problems), which have hampered the school's digitisation process. This aspect not only affects developing countries but is also related to the socioeconomic level of families with fewer resources in Western societies, that 'fourth world' invisible to the majority in developed countries [7]. Differences in the first world can, on the one hand, reproduce and even increase the formative and vital distances between the citizens of the same country, and on the other, increase the inequalities between the richest and the developing countries. In this sense, there is evidence that the coronavirus pandemic has increased educational inequality, social injustice, inequity, and with it, the digital divide [8,9].

Consequently, this research not only addresses the importance of ICT in the educational field, the training of teachers in digital competence, the digital literacy of students and their training in the technical and ethical use of technologies, but also their social impact in global citizenship and the commitment of teachers and researchers to social sustainability. In this sense, our study incorporates elements such as accessibility, equality, inclusion or health into the teaching digital competence, aspects that are inseparable from the technologies that will mark the future of humanity in the coming years [10].

*1.2. Teacher Digital Competence Training*

UNESCO [11] provides guidance on how countries should approach the development of ICT internationally to accelerate the progress of the Sustainable Development Goal (hereafter, SDG) 4: 'Ensure inclusive, equitable and quality education and promote lifelong learning opportunities for all' [12]. In this way, the demands of the digital age and the need to guarantee this right in the current information and knowledge society make it essential for teachers to master a spectrum of basic or functional digital competences and demonstrate digital leadership in the classroom to empower students in the use of ICT. At the same time, teachers are required to acquire digital skills that allow them to create and exchange digital content, communicate, collaborate, and solve problems [13]. Consequently, teacher training in digital and information literacy is essential for the acquisition and dissemination of digital skills that promote the proper use of ICT in schools [14,15]. In fact, the European Framework of Digital Competence for Educators (DigCompEdu) alludes to the need for teachers to train students in the application of digital technologies in a critical and responsible way [16].

With reference to this framework, national plans for digital competences have been designed, such as the Common Framework of Digital Competence for Teachers in the Spanish context, which values and defines the digital competences that 21st century teachers

need to develop for the improvement of their educational practice and for their professional development [17]. In addition to this European framework, there are different models that have defined teacher digital competence in an international context [18], such as the National Educational Technology Standards for Teachers (NETS-T) proposed by the International Society for Technology in Education [19]. In Spain, the National Institute of Educational Technologies and Teacher Training (Instituto Nacional de Tecnologías Educativas y de Formación del Profesorado, hereafter INTEF, Calle de Torrelaguna, 58, 28027 Madrid) is the unit of the Ministry of Education and Vocational Training (Ministerio de Educación y Formación Profesional, Calle Alcalá, 34. 28014 Madrid) of the Government of Spain that works in collaboration with administrations and educational institutions to advance towards a digitally competent education. It pursues the development of teacher digital competence through the framework (approved on 14 May 2020 by Agreement of the Education Sector Conference (Conferencia Sectorial de Educación) and published in the Resolution of 2 July 2020, of the General Directorate of Evaluation and Territorial Cooperation (Dirección General de Evaluación y Cooperación Territorial) [20]).

The Common Framework for Teacher Digital Competence is the basis of the Teacher Digital Competence Portfolio, INTEF's digital instrument for the accreditation of said competence. This Common Framework is comprised of 21 competencies. Each of these competencies comprises six levels, in which descriptors based on knowledge, skills, and attitudes are specified, becoming a key tool to detect training needs and accredit the Teacher Digital Competence [17]. Specifically, this service is aimed at improving the digital competence of teachers in the Spanish educational system through continuous self-assessment and participation in teaching, learning, and training experiences [20].

The interest of institutions in improving the digital competence of teachers affects the use made of ICT in education and its relationship with sustainable development [21]. In particular, the development of digital competence is related to social sustainability, which, together with the economy and the environment, constitute the three pillars on which the concept of sustainability is founded, in accordance with the United Nations (hereafter, UN) prescriptions [10,22]. Among the issues usually associated with social sustainability are social justice and equality, human rights, accessibility, education, or environment [23]. Consequently, actions that encourage, from teacher training, the homogeneous development of the digital competence in a community, promote a sustainable society.

*1.3. Social Sustainability in the Framework of Teacher Digital Teaching Competence*

Social sustainability has been the latest pillar of sustainability to be defined, and its conceptualisation is not yet homogeneous [10,24]. In general, social sustainability implies 'equitable access to learning and job opportunities, social mobility, social cohesion and justice, quality of life, participation, empowerment, and cultural identity based on self-confidence and a balance between innovation and tradition' [25], p. 44. Thus, a sustainable society must address access to digital competence on equal terms, since it is present in all areas of our society, and especially in the professional context.

To define social sustainability, [22] relied on its relationship with social capital, related to human capital through health and education. Social sustainability can be described as the sum of formal and informal processes, systems and networks of relationships that work together to build healthy and liveable communities [26]. Nowadays, a good part of these social networks is built through ICT. This system is based on physical and non-physical factors [27], including education, social capital, or cultural tradition [28,29]. Among these, sensitivity and respect for local traditions and heritage stand out, which are essential to strengthen informal support networks that should mediate issues related to health, education, or employment [30]. As indicators of social sustainability equality in access to key services, the authors of [10] suggest a system of cultural relations that value the positive aspects of different cultures, intergenerational equity, a system for transmitting awareness of social sustainability from one generation to the next, and a sense of community responsibility that allows this transmission system to be maintained. In short,

the development of the digital competence must coexist with this social and cultural heritage and contribute to the improvement of social cohesion and living conditions in a community.

In this context, education is understood as one of the essential ways to generate this intergenerational transmission, and thus appears as the fourth of the 17 UN SDGs [31]. From the viewpoint of social sustainability, education affects equity, both intergenerationally and within the same generation [26], and is considered as one of the key elements of sustainable societies [28]. The development of digital competence has become one of the fundamental challenges for achieving this social sustainability in education. Thus, next to the concept of social sustainability is that of digital inclusion, 'described as the promotion of motivation and capability for the use of information and communication technology in an entrepreneurial and critical way, whose objective is the development of the community and historical, political, and ethical awareness-raising' [25], p. 46. In general, the social dimension of education must be able to provide support for people to lead a dignified life and to influence their own lives and the communities in which they live [32]. An education in this framework must encompass, through ICT, the ability to analyse issues from multiple perspectives, respond quickly to changes, and foster dialogue and consensus [31,33].

Access to digital competence in educational settings affects the employability of the members of a community, and it involves not only basic training, but also lifelong training [24,25]. Education for a sustainable society is assumed to increase employment rates and thus, leads to greater economic prosperity within the community. This double perspective aims to create a better society and, at the same time, to foster and maintain employability [10]. However, the relationship between social sustainability, digital competence, and employability runs the risk of being reductionist, insofar as it can lead to understanding sustainability from a productive and commercial viewpoint [24].

In this sense, education in a sustainable society must pay special attention to educational opportunities, migratory groups, and gender diversity [24], the key factor being accessibility to digital competence, especially education, public services, or cultural and recreational infrastructure [27]. This concept is particularly relevant in developing regions, where social sustainability 'can be defined as the capability of a human to gain and maintain a decent livelihood, minimising social exclusion and improving access to social, legal, educational, and health services' [25], p. 44. It is in these environments where education in digital competence becomes an essential factor in equal opportunities and employability, both at the level of general education and continuing training.

The interest aroused in recent years on the topics developed in the conceptual framework presented in our introduction leads to the following two research questions:

**RQ1.** *What and how is research on teacher digital competence currently being addressed in scientific publications indexed in SCOPUS?*

**RQ2.** *To what extent is social sustainability integrated into current research on teacher digital competence?*

## 2. Methodology

### 2.1. Selection Process

This paper presents a review of the research literature carried out specifically on teacher digital competence and social sustainability. Consequently, we have followed a methodology typical of the systematic reviews to ensure systematisation; specifically, our research has made use of the Preferred Reporting Items for Systematic Reviews and Meta-Analyses (hereafter, PRISMA) method [34]. The development of the systematic review through PRISMA consists of four phases (Figure 1): Phase 1: Identification of the publications on the object of study housed in the SCOPUS database. SCOPUS has been selected as our main source because it is currently the database that includes a higher number of publications related to education worldwide, accomplishing quality standards such as the blinded peer review process. Phase 2: Analysis of their abstracts. Phase 3:

Full text analysis by means of the Preferred Reporting Items for Systematic review and Meta-Analysis Protocols 2015 [35]. Phase 4: Selection of publications for content analysis.

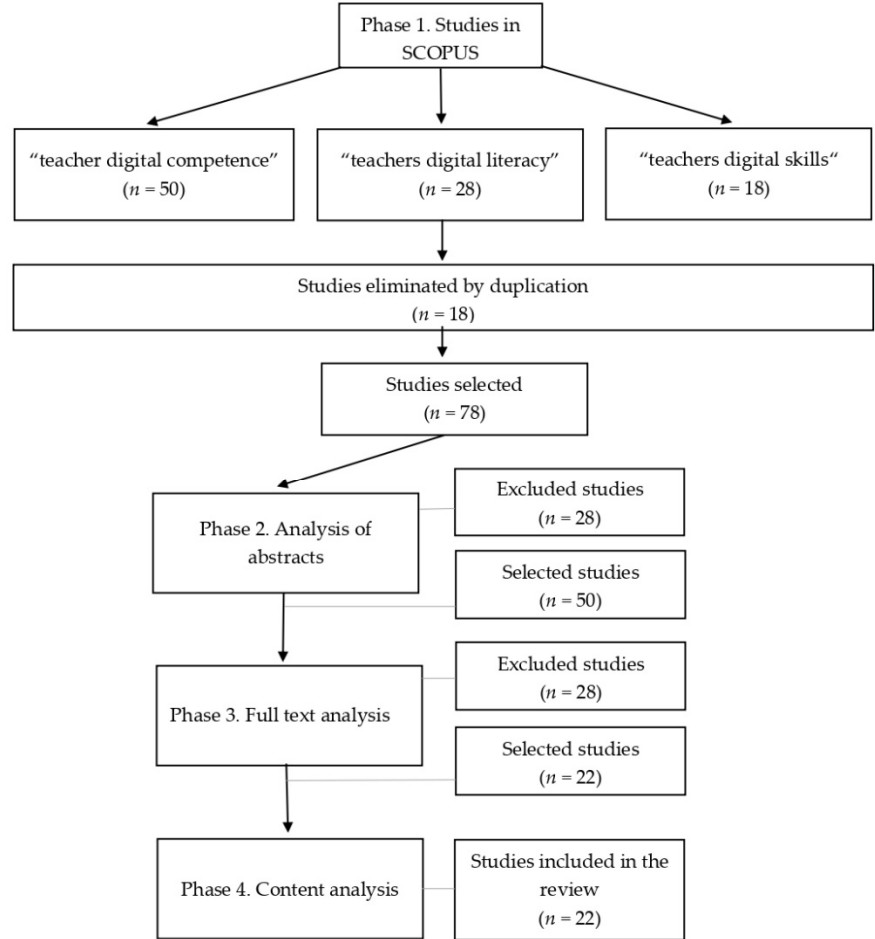

**Figure 1.** Flow diagram of the selection process of relevant studies for the systematic review.

Phase 1 was based on searching for documents in the SCOPUS database. This search was carried out considering the presence of the following terms in the article's title, abstract or the keywords as selection criteria: 'digital teaching competence', 'teachers' digital literacy' and 'teachers' digital skills'. Given the interest in knowing the entire theoretical body existing in this line, no specific search criteria were defined in relation to the type of document, language, or year of publication. Once the studies were identified, the references were exported to Mendeley, and duplicate cases were removed. This first phase yielded a total of 78 works related to the subject of interest.

Phase 2 was directed to the analysis of the abstracts. To this end, the PRISMA Checklist was used, establishing certain qualifications in the criteria, such as:

1.  The terms 'digital', 'teaching' and/or 'education' appear in the title.
2.  Research objectives or questions are delimited, including components such as participants, interventions, comparison elements, and results, focused on the teacher's digital skills.
3.  The eligibility criterion is the identification of the study characteristics that imply development in an educational environment and methodological design with a qualitative, quantitative, or mixed character.
4.  The sources of information are specified by referring to a data collection system.
5.  Reference is made to methods for assessing the risk of bias.
6.  The data analysis method is reflected as a synthesis of the results.
7.  The studies included provide results on teachers' digital competence.

8. The synthesis of the results provides a diagnosis of the level of competence.
9. Strengths and limitations of the evidence are highlighted in the discussion.
10. There is a general interpretation of the results and important implications in the discussion that incorporate a reflection on the social impact and sustainability component (inclusion, equity, well-being, intergenerational educational transfer, etc.).

The analysis of the abstracts was performed by triangulation between pairs of researchers, reaching a consensus of the score in each criterion, depending on whether it is fulfilled, with a range of total scores between 0–10 points. The papers with a total score of six or more points were selected for the next screening in Phase 3, constituting a total of 50 studies.

Phase 3 was focused on the analysis of the full text of the publications selected in the previous phase. On this occasion, the first 14 criteria of the Checklist Preferred Reporting Items for Systematic review and Meta-Analysis Protocols 2015 [35] were used, with the following adjustments:

1. The criterion on the explicit identification in the title of the terms 'digital', 'teaching' and/or 'education' is corroborated.
2. The record is identified in the document through the DOI.
3. The authors' contact details are provided, alluding to the name, institutional affiliation, and email address of all the authors of the protocol, the author's postal address is provided for correspondence.
4. If the protocol involves a correction of a protocol previously completed or published, it is identified as such, and the changes are listed; if not, the strategy to document important protocol corrections is stated.
5. The funder is identified by referring to a private or public entity.
6. There is a justification of the work, contextualized in what is already known about the subject.
7. If the objectives raise a focus on teachers, they refer to the educational context and outcomes at the level of digital competence (participants, interventions and outcomes: PIO).
8. The identification of the study characteristics that imply development in an educational setting and methodological design with a qualitative, quantitative, or mixed character, is reaffirmed as the eligibility criterion.
9. The sources of information are specified by referring to a data collection system.
10. The procedure for applying instruments and collecting data as a search strategy is described.
11. The statistical software used to manage the data is specified.
12. All variables for which data will be searched are listed and defined.
13. The analysis dimensions are delimited towards the expected results.
14. The methods used to assess the risk of bias are detailed.

Similar to the first analysis or filtering, the criteria scores were established by means of triangulation by pairs of researchers. On this occasion, the scoring range was found to be between 0–14 points, with those with a total score exceeding seven points being selected. The total number of works selected in this phase was 22.

Finally, Phase 4 consisted of the content analysis of the studies.

In addition to the use of PRISMA, this research took the fundamental principles postulated by [36] as a procedure in review works into consideration. Thus, prior to the search in SCOPUS, a critical question of interest to the scientific framework was investigated, that is, the social sustainability in the context of research on teacher digital competence, and specific research questions were posed to provide answers to the question that would guide, in a third step, the formulation of the appropriate search parameters and the determination of the inclusion and exclusion criteria of the publications to be analysed. Similarly, after the application of the PRISMA phases, in the last step, an attempt was made to consolidate and summarise the main findings while seeking the interpretation and communication of the main critical issues encountered.

### 2.2. Review Process

All studies were analysed and coded by pairs of researchers, who triangulated and agreed on the results when inquiring about their objectives, purposes, and research questions, the methodological framework (participants, context, socioeconomic level of the participants, biases that can occur in the research, methodological design, procedure, instruments, and data analysis), the results, and the conclusions that imply indicators of social sustainability. The list of works submitted for review is provided in Table 1, indicating their citation, the reference country where the studies were carried out, and the level of development of the country in question.

**Table 1.** Studies included in the review.

| Citation | Reference Country | Development Level of the Country (World Bank, 2021) |
|---|---|---|
| [37] | Turkey | Medium |
| [38] | Spain | High |
| [39] | Montenegro | Medium |
| [40] | Spain | High |
| [41] | Spain | High |
| [42] | Spain | High |
| [43] | Spain | High |
| [44] | Ghana | Low |
| [45] | Spain | High |
| [46] | Spain | High |
| [47] | Spain | High |
| | Spain | High |
| [48] | Italy | High |
| | Ecuador | Medium |
| [49] | Turkey | Medium |
| [50] | Spain | High |
| [51] | Spain | High |
| [52] | Spain | High |
| [53] | Spain | High |
| [54] | South Africa | Low |
| [55] | Spain | High |
| [56] | Italy | High |
| [57] | Spain | High |
| [58] | Ghana | Low |

## 3. Results

### 3.1. Characteristics of the Teacher Digital Competence Research Reviewed

A total of 22 articles that met the inclusion criteria indicated in the SCOPUS database were selected. As shown in Table 1, the works have been published in the last decade, specifically, in the last seven years (2015–2021). In particular, the largest volume of studies is dated between 2019 and 2020, with 19 of the 22 publications, representing 86.4% of the studies submitted for review. In 2019, eight studies (36.4%) were published, and 11 studies (50%) were published in in 2020. Thus, between 2020 and 2021, the studies exceed half the volume of the works reviewed. On the other hand, 19 of the 22 articles were developed in Europe (86.4%) and mainly with a Spanish population. In fact, 15 of the 22 studies were conducted in Spain (68.2%), compared to 31.8% that incorporate a Turkish, Italian, Ghanaian, Montenegrin, Ecuadorian, or South African population. Furthermore, only one of the studies makes a comparison between different populations. Accordingly, 16 of the articles, which represent 72.7% of the publications, contemplated environments with high developed countries, and only two were developed in more vulnerable contexts.

In relation to the objectives, purposes, and research questions, the works are classified according to whether their purpose is the design of instruments to record teacher digital competence, the measurement of it, the attention to the opinions of the teaching staff or the

students, or the performance of a comparison or evaluation of the predictive capacity of certain factors in the teacher for the use of ICT. In particular, 15 of the 22 studies (68.2%) raise the need to measure levels of teacher digital competence: four (18.2%) differentiating it by factors such as sex, age or academic experience, and two (9.1%) relating it to the teachers' strategies and aims. The rest of these 15 studies focus on analysing the levels of teacher digital competence from the viewpoint of the students (4.5%), in relation to Augmented Reality (4.5%), digital stories (4.5%), an intervention in educational robotics (4.5%), and previous training in ICT to identify teacher training needs (4.5%), social networks (4.5%), the teacher's specialty (4.5%), and to the digitisation of learning from educational policies (4.5%). For its part, four of the 22 articles (18.2%) collect teachers' perceptions about their training in digital stories (4.5%), digital literacy (4.5%), digital divide in schools (4.5%), as well as through the portfolios of teacher digital competencies (4.5%). Finally, the three remaining works are channelled towards the creation of an instrument to register the teacher digital competence from the viewpoint of the students (4.5%), the development of a comparative study in Higher Education related to the digital literacy process during COVID-19 (4.5%), and the evaluation of the teacher's ability to use ICT according to their teaching strategies and intentions (4.5%), respectively.

The studies include three different profiles of participants: students (aged 3–18), university students, and active teachers. The representation of this population in the different articles according to the level of education is illustrated in Figure 2. As can be seen, the largest volume of published work is carried out with active teachers, mainly primary education professionals. In relation to the context of the participants, its particularities are unknown in 16 of the 22 articles (72.7%). The modality of the public, subsidised, and/or private centres is only mentioned in two specific cases (9.1%). One case (4.5%) refers to the immersion in a country with a special impact from COVID-19, another (4.5%) to the belonging of the sample to centres of different geographical areas (urban, peri-urban and rural settlements), reference is made to the presence of a vulnerable context immersed in poor regions and developing countries on two occasions (9.1%), and the digital age is referred to in a generic manner in one case (4.5%). In addition, 95.5% of the publications do not take the socioeconomic level of the participants into account, and only one (4.5%) mentions that in the study area there is a representation of the economic, social, and cultural diversity of the country.

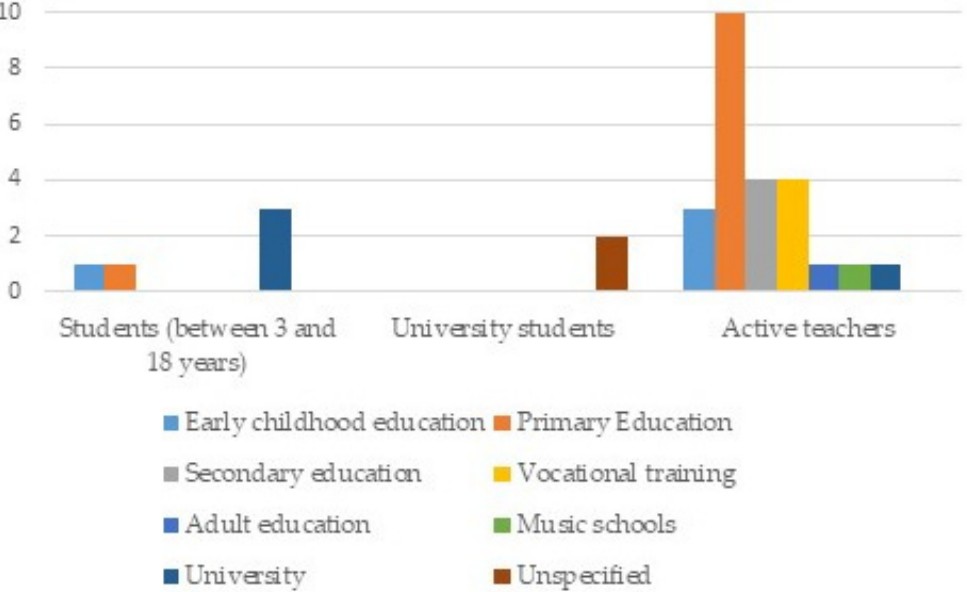

**Figure 2.** Profile of the participants of the works submitted for review.

The selection of the participants showed a certain research bias due to socioeconomic issues (motivated by the lack of accessibility to ICT in industrialised countries), as well as due to gender. Specifically, possible biases due to lack of accessibility to ICT are not considered in 11 of the 22 articles (50%), given that situations of vulnerability or problems of student access are not addressed (18.2%); the need for training teachers in the use of specific digital tools (4.5%) and the availability of digital resources in educational centres is not taken into account (22.7%). Only two publications contemplate biases due to socioeconomic issues and the consequent difficulty in accessing ICT, since the research is framed in a context where the digital divide has its origin in the lack of provision of digital resources in the centres (9.1%).

No works that explicitly take the possible existence of biases due to gender issues into account were found. However, it can be highlighted that five of the 22 articles (22.7%) state the verification of differences in competence or perception between men and women as one of their research objectives. Furthermore, one (4.5%) makes reference to lesser training in ICT received by the female participants in the study. On the contrary, the distribution of the sample by gender was not mentioned at all in two articles (9.1%).

The methodological design of the studies is mostly quantitative (18 of 22 articles; 71.8%), compared to qualitative (two of 22 articles; 9.1%) and mixed (two of 22 articles; 9.1%). In this sense, the publications refer to the exclusive or combined use of a descriptive design, that is, non-experimental, inferential, correlational, multiple cases, case study, or design in the data collection (see Figure 3). In general terms, most of the publications apply a quantitative and descriptive not experimental design (13 of 22 articles, 58.5%).

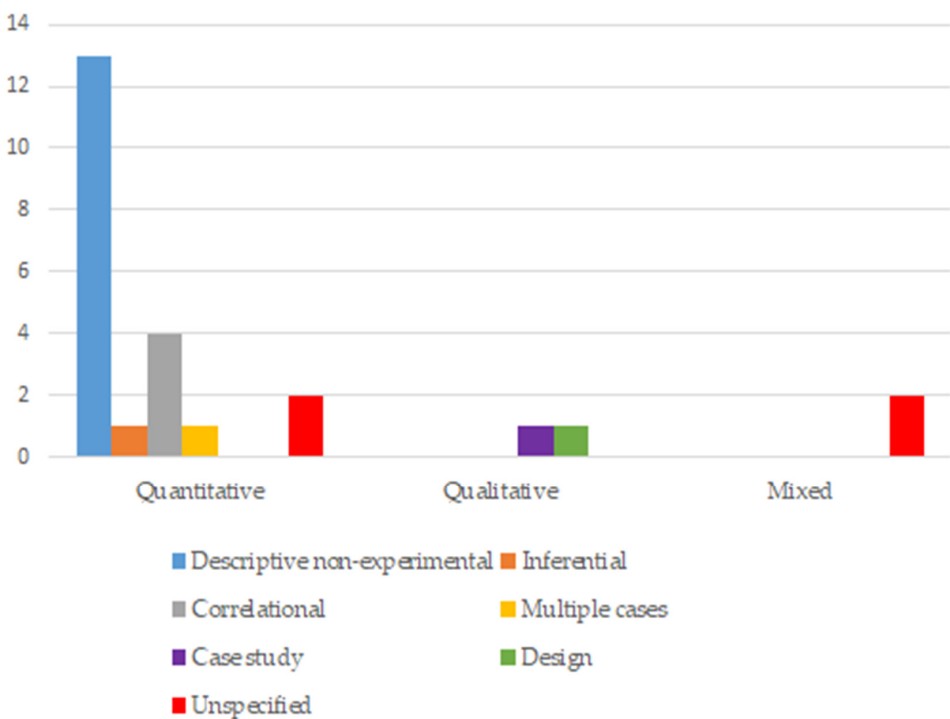

**Figure 3.** Methodological design of the publications.

Regarding the methodological procedure adopted, most (20 of 22 studies, 92.1%) are based on the administration of questionnaires; the most frequent procedure, found in ten articles (45.5%), is the elaboration of the questionnaire, its subsequent validation, and its application to the research participants. In addition to the ad hoc questionnaire made specifically for these studies, two of these 10 articles (9.1%) used other already validated questionnaires. Only questionnaires already validated or based on such were used and applied in eight of the 22 studies that constitute the analysed sample (36.4%). Only one study (4.5%) adopted a pretest–posttest design to verify the impact of a specific

training programme, accompanied by the collection of information on the opinions and perceptions of the participants in this process. Finally, a single study (4.5%) conducted different interviews with the participants as a data gathering technique.

As could be inferred from the methodological procedures described above, the works generally used the questionnaire approach. In 11 articles (50%), a validated ad hoc questionnaire composed of closed-ended questions was administered to the participants, applied electronically on eight occasions (36.4%), in person in one work (4.5%), and not specified whether it was virtual or face-to-face in two of the studies (9.1%). In another 11 articles (50%), a previously validated questionnaire with open-ended questions was used, although in two of these investigations, ad hoc questionnaires were supplemented, as indicated above, with previously validated ones. Regarding the administration of these validated questionnaires, they were applied electronically in six articles (27.3%), in person in two (9.1%), and this was not specified in three of them (13.6%). On the other hand, open-ended question questionnaires were used in four investigations (18.2%) (one being complemented with a pretest–posttest scale and an evaluation scale of the products carried out during the experience), and it was used as a complement to closed-ended questions questionnaires on two occasions (9.1%). Finally, one study (4.5%) used a semi-structured interview script as an instrument.

Consistent with the methodological procedure followed and the instruments used in general, statistical data analysis was adopted in 19 of the 22 articles (86.4%). Of these, four (18.2%) only focused on calculating descriptive statistics and the other 15 (68.2%) also included inferential statistics tests. We found a textual or content analysis of the data produced in three articles (13.6%), although it was also compatible with descriptive and inferential statistical analysis in two of these. Therefore, we only found a single article (4.5%) in which the analysis was only textual. Lastly, two investigations (9.1%) did not explicitly emphasise the data analysis procedure.

*3.2. The Presence of Social Sustainability*

Most of the studies offer an apparently homogeneous view of the society in which the research takes place. In this sense, 22.7% highlight the social differences within the sample they analyse. Although [53] argues how the typology of the educational centre can condition the differences in the development of teacher digital competence, this issue is almost absent in the sample. The most in-depth analyses come from samples belonging to developing countries, as they take into account the special conditions of their samples. For example, [54] presents the network of relationships that are woven between teachers, tribal cultures, NGOs, and the rest of the community members. However, it has not been possible to identify any study conducted in developed countries that analyses the relationship between teacher digital competence and social inequalities in the first world.

One of the recurring aspects in the studies (40.9% of the total) is the access to digital resources by teachers [38,56]. Most of the works that make reference to these aspects highlight the need for teachers, students, and schools to have the necessary resources for the development of the digitisation of education. [48,49] agree that investment in digital resources in schools does not by itself guarantee students' digital learning. The reason proposed by [48] for the ineffectiveness of the investment is the absence of digital teaching skills among the teaching staff. Difficulties in accessing digital resources are particularly acute in studies conducted in developing countries [58], especially due to the absence of telecommunication infrastructure that prevents access to the internet in educational centres.

In their theoretical framework, a good proportion of the studies analysed assume that the digital competence of teachers and the uses they make of the digital resources in the classroom automatically affects the digital competence of students. However, only 59.0% address it in their conclusions. Although this transfer is assumed as the logic of learning in a school context in most cases, some argue that the lack of digital media in schools makes this transfer difficult. Finally, the application of teacher digital competence in the classroom

turns teachers into mediators between traditional and innovative ways in the teaching and learning process [58].

A total of 36.3% of the articles raise aspects related to equity, inclusion, well-being, and intergenerational educational transfer. In particular, 18.1% of all the selected studies address equal conditions in access to digital resources, and the inequalities that exist due to issues related to gender, geographical, and economic conditions.

The authors of [58] suggest that digital literacy should be approached in the form of long-term objectives that eliminate differences between regions. Regarding the SDGs, only [38] affirms that digital competence can serve to improve indicators in the field of education. Finally, [46] concludes that there are no gender imbalances with regard to digital skills, while [52] finds notable differences in digital competence. The disparity between these results confirms the different methodological approaches of the studies; one, oriented to the method of self-administered surveys, and another, oriented to the evaluation of a sample after an intervention.

References [37,42] highlight some aspects related to intergenerational educational transfer. This transfer refers to the way in which the digital training of teachers conditions the use that their students will make of these same digital resources. When disaggregating the data according to the age of the teachers, [45] argues that the younger generations of teachers are promoting the use of ICT in the classroom. Finally, [56] suggests that teachers are aware of the relevance of ICT in the lives of the students. However, other key aspects of social sustainability, such as inclusion and well-being, have not been incorporated into any of the studies analysed.

The relationship between teacher digital competence and employability is approached differently. For instance, [37] relates the use of digital narratives not only with their application to the future teaching method of the students, but also with other areas such as marketing. Using samples in Faculties of Education, other studies relate digital competence with the professional perspectives of future teachers [52]. Reference [56] offers a vision in which government bodies promote digital literacy as an engine for social change.

## 4. Discussion

Our research has addressed the analysis of a research sample focused on the digital competence of teachers, considered from the perspective of education and social sustainability. More concretely, we have aimed to answer two research questions:

### 4.1. RQ1. What and How Is Research on Teacher Digital Competence Currently Being Addressed in Scientific Publications Indexed in SCOPUS?

In accordance with the criteria considered herein (22), the volume of articles identified highlights the need for greater quality scientific production on the topic of teacher digital competence. It is, therefore, a recent field of research, mostly developed in the last seven years and which has been gaining interest over time, mainly in countries with a high level of development in the European context.

Most of the works analysed are aimed at measuring the level of teacher digital competence from different perspectives. These works place the focus on the teaching staff, either to look for differences according to their profile (sex, age, academic experience, specialty, strategies, and intentions), in the use that they make of certain technologies (augmented reality, social networks, or educational robotics), or in the identification of needs in teacher training. Similarly, these publications coincide in integrating their sample active teachers, mostly of the compulsory stages (early childhood, primary, and secondary education), without paying explicit attention to the particularities of the context in which they work, nor to their socioeconomic level, or the possible biases in the research due to gender issues or the lack of accessibility to ICT of the participants or their students.

These studies are predominantly quantitative in nature with a non-experimental descriptive design, generally using questionnaires based on closed-ended questions administered virtually, either created ad hoc and subsequently or previously validated. This panorama highlights the interest in the development of different instruments to measure

teacher digital competence, which could be related to the innovative value of the subject of study. Furthermore, the predisposition in the dissemination of the questionnaires as a means of data collection would provide evidence of the interest in accessing large samples with greater ease and immediacy. Finally, and consistent with the instruments, the data analysis is usually based on descriptive and inferential statistics. The generic model of the articles is summarised in Figure 4.

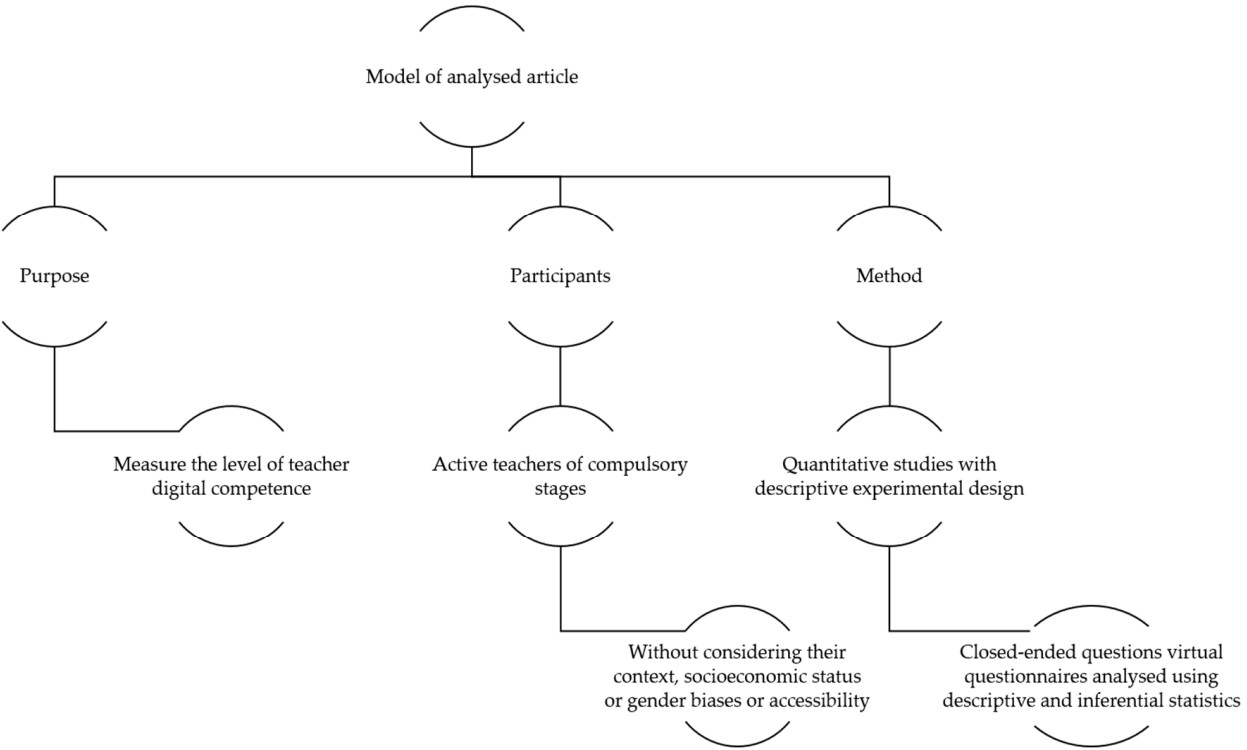

**Figure 4.** Summary of trends in research on teacher digital competence.

### 4.2. RQ2. to What Extent Is Social Sustainability Integrated into Current Research on Teacher Digital Competence?

Social sustainability requires education to make a sustained and diverse use of ICT. Therefore, education is one of the fundamental tools for the development of a digital society. Most of the studies analysed insist on highlighting the training needs of teachers to achieve the objectives of a digital society [46,47,50,53], among others. Other research, such as [43], reveals the direct relationship between the training received and the self-perception of competence in areas such as digital literacy, content creation, or security. However, this statement is not always sufficiently supported by the research results, in line with [41,50], which affirm that if a teacher does not have the necessary motivation to incorporate the ICT and the methodological tools it provides, the training will not be sufficient to achieve this end. Of the studies analysed, only [52] supports the improvement of digital competence after conducting an intervention with university students and measuring their progress in an objective manner.

After reviewing all the investigations from a social sustainability perspective, these studies tend to overlook the accessibility to digital resources of the target sample. The investigations of [39] Cortoni and Perovic (2020), [44,54,58], and [56] allow a more complex approach, to the extent that they incorporate aspects related to ICT accessibility in their sample. In this sense, they highlight the massive use of mobile phones (especially in the case of students [49]) to access the internet, compared to computers [39,44], or the deficiencies of internet networks in educational centres [39,56]. These circumstances result in teachers frequently developing their digital competence mainly in relation to the preparation of

classes, but not in their delivery. This implies a clear decrease in the presence of ICT in the teaching and learning process of students and a reduction in their digital training. Similarly, the use of laptops versus mobile phones for internet access is related to the self-perception of digital literacy in students [49] in countries such as Turkey, Ghana, and Macedonia. However, most of the research conducted in first world countries (Spain and Italy) does not take the accessibility of resources in their target population into account. From a research perspective, a notable proportion of the works present a certain bias in the data collection by not incorporating the socioeconomic and cultural aspects of the sample, especially when the use of self-reports prevents understanding the reality of the teaching practices of teachers. In this sense, the studies in which the different types of educational centres are taken into account, such as that by [53], shed light on the deficiencies in the digital training of teachers and the needs for resources.

In general, the articles present similar results, indicating that teachers do not possess the adequate digital competence necessary to approach teaching through ICT [40,53,57]. However, most of the studies do not differentiate the knowledge and use of ICT and its application to classroom teaching [40]. Similarly, the fact that many studies are based on the same theoretical and normative framework, which that includes issues related to social sustainability (such as access to resources and employability), does not guarantee that social sustainability aspects are incorporated into their analyses. The aspects that are absent in these investigations and that cause a certain bias in their research results are as follows:

1.  Heritage culture of teachers. Most of the studies focus on the current competence of teachers, without taking other factors such as their social, family, or cultural context into account. However, [49] addresses the environment of the students that constitute the sample, but this factor is systematically ignored in the case of studies conducted on active teachers. Reference [56] addresses the perception of teachers about the digital culture of their students. Only [54] mentions teacher support networks and systems in rural areas of South Africa, highlighting the weak relationships between teachers in these areas and tribal cultures and heritage.

2.  Intergenerational transmission. The studies tend to focus on the digital competence of students or teachers in educational settings. However, these studies hardly focus on the transmission of knowledge between generations (in this case, how teacher digital competence influences student learning). However, the intergenerational transmission of knowledge is one of the fundamental pillars of social sustainability in the field of education, since one of the foundations of sustainable societies is the guarantee that the learning of skills is passed from one generation to another. Most of the studies are based on the fact that, if a teacher has adequate digital competence, this will manifest itself in their teaching and will be transmitted to their students naturally. However, in a study on teacher digital competence and its use during the lockdowns caused by the COVID-19 pandemic in Spain, Ecuador, and Italy, [48] demonstrates that the perception of ICT use by teachers among students is very negative. This highlights the limitations of teachers to perceive digital competence as a transversal element in the teaching and learning process in education.

3.  Access to resources. One of the foundations of a sustainable society is access to the resources necessary for its development. However, the studies analysed only address this aspect of teacher digital competence when the research is conducted in underdeveloped or developing countries. In the same way, only when the methodologies used move away from the self-administered questionnaires distributed online, without differentiating the type of centres in the sample, do the needs and deficiencies in access to resources present in the first world arise [53,56]. This circumstance leads us to rethink the methodological approaches that exclude accessibility to technological resources and draw a somewhat contradictory map depending on the research perspective, especially in studies oriented towards industrialised countries.

4.  Gender equality. The studies present varied results regarding gender equality in relation to the digital competence of teachers and students. On the one hand, research

focused on the gender perspective in relation to teacher digital competence does not find significant differences in this area [46,55]. On the other hand, [49] found slight differences in students' digital competence depending on the cultural level of their parents. Hence, the results of the intervention by [52] are particularly interesting, since they indicate a clear difference between the digital competence of students to the detriment of men. The differences between these two studies are due to the fact that the former is based on student self-perceptions, while [52] assesses a classroom intervention. The disparity in these results must be attributed to the research methodology used, which highlights the possible gender bias in most of the research carried out so far.

5.  Employability. One of the requirements of education in a sustainable society is that it is capable of providing training that facilitates the employability of its members and, therefore, the improvement of their living conditions and that of their descendants, as well as the general enrichment of society. However, the relationship between the digital competence of teachers and their professional projection or employability is overlooked in the analyses of the studies reviewed. It is possible that this is a consequence of the very nature of the studies, focused largely on the measurement of the subjects' self-perception in a series of specific parameters, and in the absence of longitudinal studies or contrasted with the levels of employability. This aspect has become one of the lines of work that can strengthen the development of digital competence among future teachers.

## 5. Conclusions

Taking the analysis conducted and the gaps detected in relation to social sustainability into account, it would be convenient to open lines of research on teacher digital competence explicitly related to the social and economic repercussions of the inclusion of ICT in the educational field, considering the relevance of equal access, the reduction in the digital divide, heritage culture and foresight in investigations of social biases such as gender or environmental impact. This is even more relevant in the critical moments that society is currently experiencing due to the emergency situation in the use of ICT due to the COVID-19 pandemic. The consequences of this pandemic could impact research on teacher digital competence in the next few years after our review. Addressing the gaps in the research on social sustainability and the use of ICT would contribute to the development of a sustainable society that is oriented towards social and intergenerational justice. Therefore, future research in this field should attend to these circumstances. At the same time, it should take into account Web of Science or the Nordic Index as data sources.

**Author Contributions:** Conceptualization, A.M.D.l.C., A.P.-C., M.Á.G.-R. and F.G.-S.; methodology, A.M.D.l.C., A.P.-C., M.Á.G.-R. and F.G.-S.; software, A.M.D.l.C., A.P.-C., M.Á.G.-R. and F.G.-S.; validation, A.M.D.l.C., A.P.-C., M.Á.G.-R. and F.G.-S.; formal analysis, A.M.D.l.C., A.P.-C., M.Á.G.-R. and F.G.-S.; investigation, A.M.D.l.C., A.P.-C., M.Á.G.-R. and F.G.-S.; resources, A.M.D.l.C., A.P.-C., M.Á.G.-R. and F.G.-S.; data curation, A.M.D.l.C., A.P.-C., M.Á.G.-R. and F.G.-S.; writing—original draft preparation, A.M.D.l.C., A.P.-C., M.Á.G.-R. and F.G.-S.; writing—review and editing, A.M.D.l.C., A.P.-C., M.Á.G.-R. and F.G.-S.; visualization, A.M.D.l.C., A.P.-C., M.Á.G.-R. and F.G.-S.; supervision, A.M.D.l.C., A.P.-C., M.Á.G.-R. and F.G.-S.; project administration, A.P.-C.; funding acquisition, A.P.-C. All authors have read and agreed to the published version of the manuscript.

**Funding:** This paper is part of the I+D+I project PID2019-104557GB-I00, funded by MCIN/AEI/10.13039/501100011033/.

**Institutional Review Board Statement:** Not applicable.

**Informed Consent Statement:** Not applicable.

**Conflicts of Interest:** The authors declare no conflict of interest.

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
