# Peer review of "Understanding Teacher Digital Competence in the Framework of Social Sustainability: A Systematic Review"

_sustainability, doi:10.3390/su132313283_

Round 1

Reviewer 1 Report

The article is well structured. The selection of literature is relevant, perhaps a Web of Science database could be included as an extension of the work in the future.
It might also be interesting to see what the Nordic countries publish within their Nordic Index.

Author Response

Identificador: Sustainability – 1451521

 We are truly thankful for the reviewers’ careful reading of our article. We have tried to incorporate the suggestions made, which undoubtedly have contributed to improve the research performed. All the changes in the text are highlighted in blue throughout the article’s text. Besides, in the following tables we provide a response to the comments and suggestions of each reviewer.

Reviewer 1

The article is well structured.

Reviewer’s comments

Authors’ answer

1

The selection of literature is relevant, perhaps a Web of Science database could be included as an extension of the work in the future.

Attending to the reviewer’s suggestion, we have included it in the “Conclusions” section, which now reads:

“Therefore, future research in this field should attend to these circumstances. At the same time, it should take into account Web of Science or the Nordic Index as data sources.”

2

It might also be interesting to see what the Nordic countries publish within their Nordic Index.

We have included the reviewer’s suggestion in the “Conclusions” section, which now reads:

“Therefore, future research in this field should attend to these circumstances. At the same time, it should take into account Web of Science or the Nordic Index as data sources.”

Reviewer 2 Report

It is more correct to use ICT without the "S" at the end (ICTs). Otherwise the methodology is adequate and the work is very complete and correct.

In the list of bibliographical references, references 7 and 10 lack the date, some of them are not in bold. 

Author Response

Identificador: Sustainability – 1451521

We are truly thankful for the reviewers’ careful reading of our article. We have tried to incorporate the suggestions made, which undoubtedly have contributed to improve the research performed. All the changes in the text are highlighted in blue throughout the article’s text. Besides, in the following tables we provide a response to the comments and suggestions of each reviewer.

Reviewer 2

Otherwise the methodology is adequate and the work is very complete and correct.

Reviewer’s comments

Authors’ answer

1

It is more correct to use ICT without the "S" at the end (ICTs).

It has been reviewed and amended throughout the text.

2

In the list of bibliographical references, references 7 and 10 lack the date, some of them are not in bold.

We have checked the references and we have amended the mistakes noted by the reviewer.

Reviewer 3 Report

The authors in this paper present a systematic review of research on teacher digital competence and social sustainability with a review of 22 studies indexed in SCOPUS. Authors suggest that most research is intended to measure the digital competence level of teachers, usually in compulsory stages of the educational system and through  quantitative studies based on virtual questionnaires comprised of closed-ended questions missing social sustainability and justice

This is an interesting and timely piece of research, some comments to be addressed are:

  • Covid19 is introduced as a driver of social practices in the abstract and introduction, but how does that affect the LR?
  • The authors include SCOPUS as the only database, but the reason is not clear to me.
  • The Common Framework for Teacher Digital Competence is comprised of 21 competencies how that is reflected in the review?
  • How is each author work in each of the phases detailed? (2.2 said something but more detail is needed)
  • Criteria for the searches on page 6 would benefit from a table
  • Table 1 should include more information related to the research questions, for example, and extend to other aspects, which of them are offering qualitative methods and which quantitative? Which ones raise aspects related to equity, inclusion, well-being? That needs to be identifiable in the review
  • Figures 1 and 2 are not helpful for the reader, alternative visualisation should be considered
  • Results and discussion should be answering the research questions more than the conclusions that should provide a takeaway message and limitations and future work. Reconsider therefore dividing results and discussion.

This is a good piece of research I think it will benefit if the presentation flow is changed following these recommendations

Author Response

Identificador: Sustainability – 1451521

We are truly thankful for the reviewers’ careful reading of our article. We have tried to incorporate the suggestions made, which undoubtedly have contributed to improve the research performed. All the changes in the text are highlighted in blue throughout the article’s text. Besides, in the following tables we provide a response to the comments and suggestions of each reviewer.

Reviewer 3

This is a good piece of research I think it will benefit if the presentation flow is changed following these recommendations

The authors in this paper present a systematic review of research on teacher digital competence and social sustainability with a review of 22 studies indexed in SCOPUS. Authors suggest that most research is intended to measure the digital competence level of teachers, usually in compulsory stages of the educational system and through quantitative studies based on virtual questionnaires comprised of closed-ended questions missing social sustainability and justice

This is an interesting and timely piece of research, some comments to be addressed are:

Reviewer’s comments

Authors’ answer

1

Covid19 is introduced as a driver of social practices in the abstract and introduction, but how does that affect the LR?

The articles selected for their review were not influenced by COVID-19 pancemia. For this reason, COVID-19 is mentioned in our research. This actuality is one of the reasons for mentioning COVID-19 in the text, due to the necessity of including it explicitly in future research.

Attending to the reviewer’s suggestion, we have included a brief mention in the “Conclusions” section:

“This is even more relevant in the critical moments that society is currently experiencing due to the emergency situation in the use of ICT due to the COVID-19 pandemic. The consequences of this pandemic could impact research on teacher digital competence in the next few years after our review.”

2

The authors include SCOPUS as the only database, but the reason is not clear to me.

According to this suggestion, we have specified the reasons for choosing SCOPUS as our main source in section 2.1 (Selection process), which now reads:

“SCOPUS has been selected as our main source because it is currently the database that includes a higher number of publications related to education worldwide, accomplishing quality standards  such as the blinded peer review process.”

3

The Common Framework for Teacher Digital Competence is comprised of 21 competencies how that is reflected in the review?

The review presented in this article does not aim to analyse Teacher Digital Competence itself, but to approach it as related to social sustainability. In Section 1.3 (Social sustainability in the framework of teacher digital competence), we delve in this relation, always taking into account how the five fields and 21 competences of the Common Framework dialogue with specific social sustainability issues. The main objective of these competencies, namely the development of permanent lifelong learning skills, is present in our perspective on social sustainability. In this sense, our research highlights the gap between the acquisition of lifelong learning skills and social sustainability in the analysed publications. For this reason, understanding the ways in which the research in our sample faces teacher digital competence  leads us to a more accurate knowledge of their limitations regarding lifelong learning skills.

4

How is each author work in each of the phases detailed? (2.2 said something but more detail is needed)

The authors of this article have worked equally in each of the phases of this research. For this reason, the particular contributions have not been specified in Section 2.2 (Review process). As stated in the final section “Author contributions”, all the researchers have participated in the research’s method, analysis and investigation.

5

Criteria for the searches on page 6 would benefit from a table

Attending this suggestion, the authors have tried to include the criteria in a table. However, it has been impossible to present all the details in a visually clear way. In order to avoid duplications with the information contained in the article’s text, and considering the already extant flow diagram of the research process (Figure 1), we have decided to keep the current exposition of information throughout the text. Thus, although the visual display may be less structured, the reader keeps all the details regarding our criteria and research.

6

Table 1 should include more information related to the research questions, for example, and extend to other aspects, which of them are offering qualitative methods and which quantitative? Which ones raise aspects related to equity, inclusion, well-being? That needs to be identifiable in the review

We haven’t included the research design in Table 1 because it is presented in Figure 3, with more details about the analysis performed in the publications analysed. Other aspects related with the topics of the research have not been included, as none of the articles analysed focused directly on the topics suggested by the reviewer (equity, inclusion, well-being, or similar social sustainability linked topics). The way in which these issues are addressed in the publications reviewed is widely described in Section 3.2.

7

Figures 1 and 2 are not helpful for the reader, alternative visualisation should be considered

As Figure 1 is a flow diagram, we understand that this comment affects Figures 2 and 3. Attending to the suggestion made, we have modified these two figures:

-Figure 2. We have chosen a plain design, without effets, and we have improved the display quality.

-Figure 3. We have modified the type of graph, thus being more consistent with Figure 2. We have modified the scale and the display’s colours.

We hope the current display makes Figures 2 and 3 clearer to the readers.

8

Results and discussion should be answering the research questions more than the conclusions that should provide a takeaway message and limitations and future work. Reconsider therefore dividing results and discussion.

We have split the Results and Discussion section, as suggested. As a consequence, the current structure offers a clearer display of our findings.

Round 2

Reviewer 3 Report

Thanks I am glad to see that the reviewers have addressed my suggestions. I think it is ready for publication.